# Signature of anyonic statistics in the integer quantum Hall regime

P. Glidic[1], I. Petkovic [1] ✉, C. Piquard[1], A. Aassime[1], A. Cavanna [1], Y. Jin[1], U. Gennser[1], C. Mora[2], D. Kovrizhin[3], A. Anthore[1,4] & F. Pierre[1] ✉

Anyons are exotic low-dimensional quasiparticles whose unconventional quantum statistics extend the binary particle division into fermions and bosons. The fractional quantum Hall regime provides a natural host, with the first convincing anyon signatures recently observed through interferometry and cross-correlations of colliding beams. However, the fractional regime is rife with experimental complications, such as an anomalous tunneling density of states, which impede the manipulation of anyons. Here we show experimentally that the canonical integer quantum Hall regime can provide a robust anyon platform. Exploiting the Coulomb interaction between two copropagating quantum Hall channels, an electron injected into one channel splits into two fractional charges behaving as abelian anyons. Their unconventional statistics is revealed by negative cross-correlations between dilute quasiparticle beams. Similarly to fractional quantum Hall observations, we show that the negative signal stems from a time-domain braiding process, here involving the incident fractional quasiparticles and spontaneously generated electron-hole pairs. Beyond the dilute limit, a theoretical understanding is achieved via the edge magnetoplasmon description of interacting integer quantum Hall channels. Our findings establish that, counter-intuitively, the integer quantum Hall regime provides a platform of choice for exploring and manipulating quasiparticles with fractional quantum statistics.

Integer and fractional quantum Hall effects[1] are thought of as fundamentally separate. The main features of the integer quantum Hall (IQH) states are well described within the single-particle fermionic picture[1–4]. In contrast, fractional quantum Hall (FQH) states inherently stem from strong Coulomb interactions, giving rise to anyons - composite quasiparticles that carry a fractional charge and exhibit anyonic exchange statistics[5,6]. Abelian anyons acquire a phase upon exchange, whereas non-abelian anyons undergo a deeper transformation into different states[7]. Demonstrating the exchange statistics of anyons, in particular non-abelian, is a crucial stepping stone towards realizing topological quantum computing[8].

Anyonic exchange statistics can be revealed via interferometry, whereby anyons along the edge move around those in the bulk, and acquire a braiding (double exchange) phase[6,9–11]. An alternative probe, not requiring involved heterostructures with built-in screening, is provided by a mixing process at an 'analyzer' quantum point contact (QPC)[12]. If the impinging quasiparticle beams are dilute (Poissonian), the outgoing current cross-correlations carry a signature of anyonic statistics[12–14]. These dilute beams are created upstream by sources typically realized by voltage-biased QPCs set in the tunneling regime. The signature of anyonic statistics becomes particularly straightforward with two symmetric sources, a configuration often referred to as

[1]Université Paris-Saclay, CNRS, Centre de Nanosciences et de Nanotechnologies, 91120 Palaiseau, France. [2]Université Paris Cité, CNRS, Laboratoire Matériaux et Phénomènes Quantiques, F-75013 Paris, France. [3]CY Cergy Paris Université, CNRS, Laboratoire de Physique Théorique et Modélisation, Cergy-Pontoise F-95302, France. [4]Université Paris Cité, CNRS, Centre de Nanosciences et de Nanotechnologies, F-91120 Palaiseau, France. ✉e-mail: ivana.petkovic@c2n.upsaclay.fr; frederic.pierre@cnrs.fr

a 'collider'[14]. In this simple case, the cross-correlations for free fermions vanish, whereas a negative signal is considered to be a strong marker of anyonic statistics[6,14,15]. Such negative current cross-correlation signatures of anyons at filling factors $\nu = 1/3$ and $2/5$ have recently been demonstrated[13,16-18].

However, FQH states present complications that impede the analysis and further manipulation of anyons: the edge structure is often undetermined between several alternatives[19], the tunneling density of states generally presents an anomalous voltage dependence[20], and the decoherence along the edge appears to be very strong[9-11]. A promising alternative path is provided by the insight that fractional charges propagating along the edges of IQH states should also behave as anyons[21-23], although they are not topologically protected, unlike fractional bulk quasiparticles. Indeed, the exchange phase of two quasiparticles of charge $e^*$ propagating along an integer quantum Hall channel[22] is $\pi(e^*/e)^2$. This exchange phase can be linked to a dynamical Aharonov-Bohm effect[24]. In practice, such IQH anyons could be obtained e.g. by driving the edge channel with a narrow voltage pulse, from the charge fractionalization across a Coulomb island, or by exploiting the intrinsic Coulomb coupling between copropagating edge channels[22,23,25,26]. The present work proposes and implements the latter strategy, and demonstrates the anyonic character of the resulting fractional charges from the emergence of negative cross-correlations.

We focus on the filling factor $\nu = 2$, which has two copropagating edge channels and constitutes the most simple, canonical, and robust IQH state with interacting channels. The edge physics is well described by a chiral Luttinger model involving two one-dimensional channels with a linear dispersion relation and short-range Coulomb interactions[20,27-30]. This theory, which has been successful in explaining many experimental findings such as multiple lobes in a Mach-Zehnder interferometer[31-33], spin-charge separation[34,35] or noise measurements[36], reformulates interacting fermionic edge states as two free edge magnetoplasmon (EMP) modes via bosonization. In the limit of weak inter-channel coupling, each EMP mode is localized in one different channel and the system can be mapped back into the free electron picture. In contrast, at strong coupling, the two EMP modes are fully delocalized between the two quantum Hall channels and correspond to a charge mode, with identical charge density fluctuations on both channels, and a neutral mode, with opposite density fluctuations[21,26,37]. Experimentally, typical Al(Ga)As devices at $\nu = 2$ often appear to be close to the strong coupling regime[34,35,38].

Here we exploit such an inter-channel distribution of EMPs at strong coupling to split electrons into fractional charges, similar to the theoretical proposal in ref. 26. We start by injecting electrons into a single edge channel with a voltage-biased QPC. Then, downstream from the QPC, each injected electron progressively splits into two wave-packets. Assuming strong coupling, one is solely built upon charge EMPs propagating at velocity $v_c$, and the other is constructed from neutral EMPs and has a slower velocity $v_n$. If we consider separately the quantum Hall channel where the electron is injected, both wave-packets carry a fractional charge of $e/2$, whereas in the other channels they have opposite charges $\pm e/2$ (see Fig. 1a)[26,39]. Such fractional wave-packets propagate non-dispersively. Considered individually, they are predicted to behave as abelian anyons with non-trivial exchange phase[6,21-23].

To experimentally address the anyon character of fractional charges propagating along integer quantum Hall channels, we measure the current cross-correlations at the output of a 'collider' in the stationary regime (see Fig. 1a, c). As for the fractional quantum Hall version of the device[14], direct anyon collisions are very rare and can be ignored in the relevant dilute beam limit[15,23]. The cross-correlation signal stems instead from a braiding in the time-domain between incident anyons and particle-hole pairs spontaneously excited at the QPC[12,13,15,22,23,40-42], as illustrated in Fig. 1b. At integer filling factors, the pairs are always formed of fermionic particles (electrons and holes), whereas in the fractional quantum Hall regime they can consist of anyons. Therefore the time-braiding considered here takes place between two different types of quasiparticles, of fractional and integer charges. The braiding (double exchange) phase $2\theta$ acquired in such heterogeneous cases characterizes the so-called mutual quantum statistics. It is predicted to take the fractional value of $2\theta = \pi$ (compared to $0 \pmod{2\pi}$ for the braiding of fermions or bosons). Note that while the braiding mechanism is equally relevant for a single incident beam of fractional quasiparticles or for two symmetric beams, the latter 'collider' setup allows for a qualitative test of the unconventional anyon character from the mere emergence of non-zero cross-correlations at the output[14,15,41,42].

The sample, shown in Fig. 1c, is nanostructured from an Al(Ga)As heterostructure and measured at 11 mK and 5.2 T. It consists of two source QPCs (metallic split gates colored red) located at a nominal distance $d = 3.1\,\mu m$ from the central 'analyzer' QPC (yellow gates). If not stated otherwise, all QPCs are set to partially (fully) reflect the outer (inner) edge channel, and the analyzer QPC is tuned to an outer edge channel transmission probability $\tau_c \simeq 0.5$. A negative voltage is also applied to the non-colored gates to reflect the edge channels at all times, as schematically depicted. Low-frequency current auto-correlations $\langle \delta I_1^2 \rangle$ and $\langle \delta I_2^2 \rangle$ on the left and right side, respectively, and cross-correlations $\langle \delta I_1 \delta I_2 \rangle$ across the analyzer QPC are measured

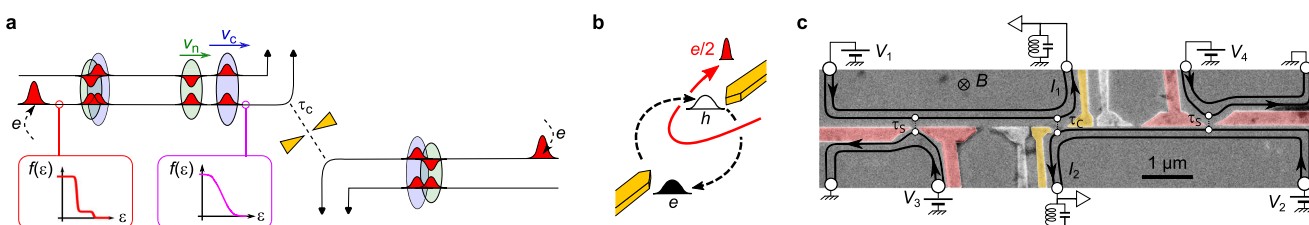

**Fig. 1 | Experimental setup. a** In the presence of two strongly coupled quantum Hall channels at $\nu = 2$, tunneling electrons $e$ (individual red wave-packets) progressively split into two pairs (circled). The fast 'charge' pair (blue background) consists of two copropagating $e/2$ wave-packets, one in each channel, whereas the slow 'neutral' pair (green background) consists of opposite $\pm e/2$ charges. The fractionalized $e/2$ charges propagate toward a central QPC (yellow split gates) of transmission $\tau_c$, used to investigate their quantum statistics from the outgoing current cross-correlations. The strong coupling regime and the degree of fractionalization at the level of the central QPC are established separately through the evolution of the electron energy distribution function $f(\varepsilon)$ from a non-equilibrium double step (red inset) to a smoother function (magenta inset). **b** Illustration of the time-braiding mechanism, whereby an impinging fractionalized $e/2$ charge (red) braids with an electron-hole pair (black) spontaneously excited at the central QPC. **c** E-beam micrograph of the sample. The two copropagating edge channels are drawn as black lines with arrows indicating the chirality. The aluminum gates used to form the QPCs by field effect are highlighted in false colors (sources in red, central analyzer in yellow). A negative voltage is applied to the non-colored gates to reflect the edge channels at all times. Tunneling at the sources is controlled by the applied dc voltages $V_{1,2,3,4}$ and through their gate-controlled transmission probability $\tau_s$.

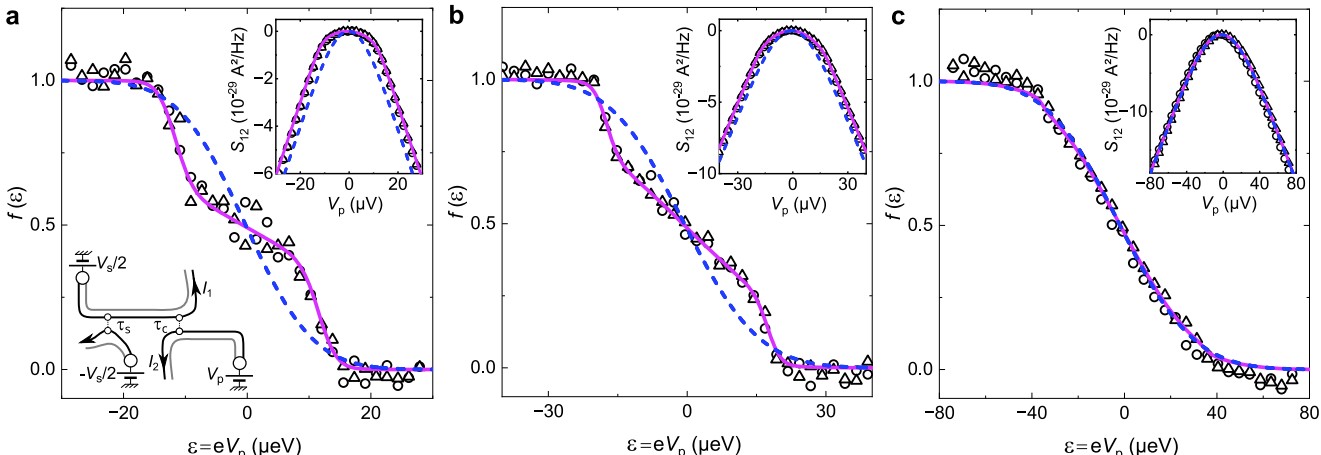

**Fig. 2 | Spectroscopy of the electron energy distribution $f(\varepsilon)$.** The shape of $f(\varepsilon)$ reflects the inter-channel coupling regime and informs on the conditions for a complete charge fractionalization at the central QPC. One source is voltage biased at $V_s$, here with $\tau_s \approx 0.5$, and the same probe voltage $V_p$ is applied across the other one (see schematic in **a**). Circles and triangles show data points with the voltage biased source QPC on the left and right side, respectively. Purple continuous lines and blue dashed lines represent exact theoretical predictions in the strong coupling regime for a time delay between charge and neutral pairs of $\delta t = 64$ ps and $\infty$, respectively (see Supplementary Information). Insets: Cross-correlations $S_{12}$ versus probe voltage $V_p$. Main panels: $f(\varepsilon)$ obtained by differentiation of $S_{12}$, see Eq. (1), with $\tau_c \simeq 0.5$. **a**–**c**: Data and theory at $T \simeq 11$ mK for a source voltage $V_s = 23\,\mu$V, $35\,\mu$V, and $70\,\mu$V, respectively.

simultaneously. In the following, the excess noises are denoted $S_{ij} \equiv \langle \delta I_i \delta I_j \rangle - \langle \delta I_i \delta I_j \rangle (V_{1,2,3,4} = 0)$ with $i, j \in \{1, 2\}$.

## Results
### Electron fractionalization

First, we need to ensure that the device is in the strong coupling regime and to determine under which conditions the tunneling electrons are fractionalized into well-separated $e/2$ charges at the analyzer QPC.

Our straightforward approach is to inject energy into one edge channel at a source and to probe the energy redistribution at the analyzer[43]. Indeed, the fractionalization of the tunneling electron coincides with the emergence of charge pulses in the other, copropagating edge channel and, consequently, with a transfer of energy. Furthermore, the full EMP delocalization between both channels, which is specific to the strong coupling limit, also translates into an equal redistribution of energy between the channels at long distances[44].

In this measurement, only one source QPC is used together with the analyzer. We inject energy into the outer channel by applying a constant dc voltage bias $V_s$ across the source QPC (e.g., $V_1 = V_s/2$ and $V_3 = -V_s/2$ for the left source, Fig. 1c). The resulting electron energy distribution immediately downstream of the injection point $f_{inj}$ takes the shape of a double step (red inset in Fig. 1a), where $f_{inj}(\varepsilon) = \tau_s f_{FD}(\varepsilon + eV_s/2) + (1 - \tau_s)f_{FD}(\varepsilon - eV_s/2)$, with $\tau_s$ the transmission probability of outer channel electrons across the source QPC, and $f_{FD}$ the Fermi-Dirac distribution.

The electron energy distribution spectroscopy at the analyzer is performed in the out-of-equilibrium outer edge channel by measuring the cross-correlations $S_{12}$ vs the probe voltage $V_p$ that controls the electrochemical potential of the equilibrium edge channel on the other side (e.g., $V_2 = V_4 = V_p$). The latter's cold Fermi distribution acts as a step filter[45–47], up to a $k_B T \approx 0.1\,\mu$eV rounding. The probed out-of-equilibrium electron energy distributions $f$ displayed in Fig. 2 are computed from the measured $S_{12}$ (inset) using[45,48]:

$$f(\varepsilon = eV_p) \equiv \frac{1}{2}\left(1 + \frac{h}{2e^2 \tau_c(1 - \tau_c)}\frac{\partial S_{12}(V_p)}{e\partial V_p}\right). \quad (1)$$

The three panels in Fig. 2 show the evolution of $f$ with the source bias voltage $V_s$, at $\tau_s \simeq \tau_c \simeq 0.5$. Whereas at low $V_s = 23\,\mu$V (panel a) $f$ remains close to a double-step function, we observe a marked relaxation

towards an intermediate shape at $V_s = 35\,\mu$V (panel b) and, at high bias $V_s = 70\,\mu$V (panel c), $f$ takes the shape of a broad single step closely matching the long-distance prediction for the strong coupling limit (blue dashed lines). This last observation establishes that the present device is in the strong coupling limit. Furthermore, since the data for $70\,\mu$V agrees well with the long-distance prediction, this indicates that the fractionalized $e/2$ wave-packets are already well-separated at the analyzer for a bias $V_s$ of $70\,\mu$V. The observation at $35\,\mu$V of a different distribution function, still showing remnants of a double step, indicates an incomplete separation of the wave-packets up to this voltage. Therefore the separation into two $e/2$ wave-packets occurs for a source voltage bias within the range of $35\,\mu$V and $70\,\mu$V. In that case, the wave-packet time-width $h/eV_s$[49] is smaller than the time delay between the arrival of fractionalized $e/2$ charges at the analyzer QPC $\delta t = d/v_n - d/v_c$.

Further evidence of the good theoretical description of the device is provided by the quality of the quantitative comparison between the data and the exact calculations of $f$ at finite distance (purple continuous lines). These predictions were obtained by an extension of the theory involving a subsequent refermionization of the bosonized Hamiltonian, which enables a full access to the cross-correlations and out-of-equilibrium electron distributions (see Supplementary Information). The only fitting parameter is the time delay $\delta t$. Here it is fixed to $\delta t = 64$ ps, and its associated effective velocity $d/\delta t = 5 \times 10^4$ m. s$^{-1}$ is comparable to EMP velocity measurements in similar samples[35]. In addition to the Supplementary Fig. 5 showing a comparison at additional intermediate voltages, see also Supplementary Fig. 6 for measurements with a dilute quasiparticle beam, and Supplementary Fig. 7 for a (less-controlled) power injection in the inner edge channel.

### Negative cross-correlation signature of anyon statistics

We now turn to the cross-correlation investigation of the fractional mutual braiding statistics between $e/2$ edge quasiparticles and electrons. Figure 3 displays the central measurement of cross-correlations in the configuration of two sources injecting symmetric dilute beams toward the analyzer. The source QPCs are biased at a voltage $V_s$ equally distributed on the two inputs ($V_{1,2} = -V_{3,4} = V_s/2$, Fig. 1c), and set to $V_s = 70\,\mu$V, previously established to correspond to the full fractionalization of the quasiparticles entering the analyzer. We set both source QPCs either to a transmission $\tau_s \approx 0.05$ corresponding to a dilute beam of electrons, or to $\tau_s \approx 0.95$ for a dilute beam of holes (Fig. 3a).

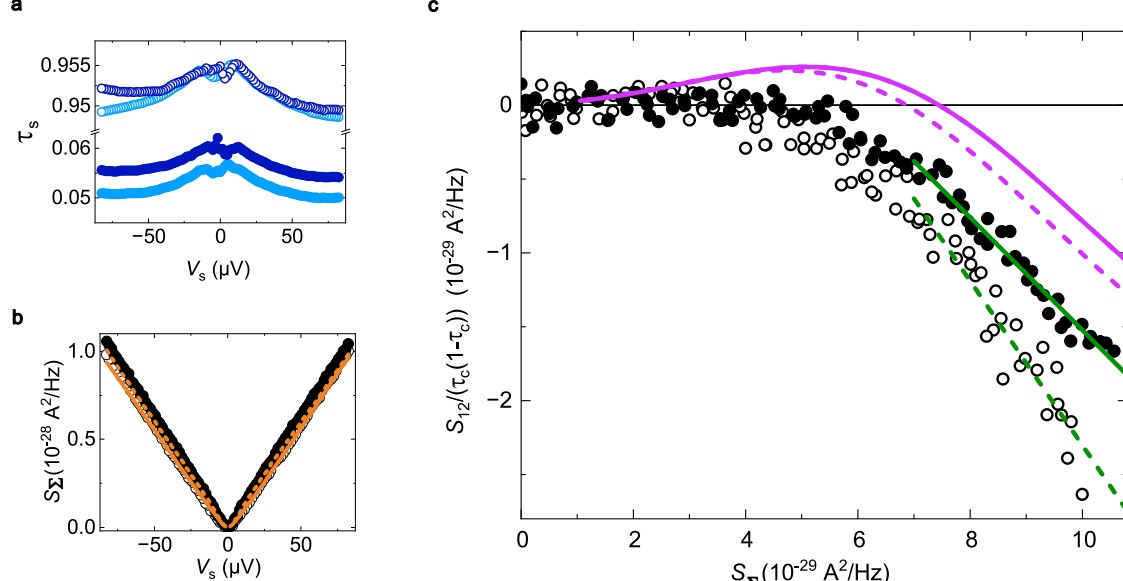

**Fig. 3 | Cross-correlation signature of fractional statistics with symmetric dilute beams. a** Measured left/right source QPC dc transmission as a function of bias voltage, shown in light/dark blue, respectively. **b** Sum of sources' shot noise $S_\Sigma$ vs source bias voltage $V_s$. The orange lines display Eq. (3) with $T = 11$ mK, the independently measured temperature. **c** Measured excess shot noise $S_{12}/(\tau_c(1 - \tau_c))$ as a function of source shot noise $S_\Sigma$ for a small source QPC transmission $\tau_s = 0.05/$ 0.95 (full/empty dots respectively). The purple lines display the strong inter-channel coupling prediction for $\delta t = 64$ ps. The green lines denotes the slope, i.e., the Fano factor (see Main text), yielding $P \simeq -0.38/-0.56$ for $\tau_s = 0.05/0.95$ respectively. In all panels full/open circles and solid/dashed lines denote $\tau_s = 0.05/0.95$, respectively.

The relevant parameter to investigate the cross-correlation signature of anyonic statistics is the generalized Fano factor[14]

$$P \equiv \frac{S_{12}}{\tau_c(1 - \tau_c)S_\Sigma}, \tag{2}$$

where $\tau_c$ is the analyzer transmission, and $S_\Sigma$ is the sum of the current noises emitted from the two source QPCs. $P$ carries information on the braiding statistics, with a high bias voltage limit that depends on the braiding phase[14,41,42]. If the (mutual) braiding statistics of free quasiparticles is trivial, such as for fermions or bosons, then $P$ is zero, whereas it is non-zero otherwise. The mere observation of a non-zero $P$ therefore provides a qualitative signature of an unconventional braiding statistics. However, we stress the importance of unambiguously establishing the underlying theoretical description of the system. For instance, negative cross-correlations could also be obtained with fermions, by phenomenologically introducing an ad hoc redistribution of energy (see Supplementary Information). Here we have established the suitability of the fractionalized charge picture through electron energy distribution spectroscopy, by comparing the observed bias voltage evolution of the energy distribution with the quantitative predictions of this model (see Fig. 2 and Supplementary Fig. 5). In practice, $P$ is extracted from the slope of $S_{12}/(\tau_c(1 - \tau_c))$ vs $S_\Sigma$, as shown in Fig. 3c (green lines). Note that the measurements of $S_\Sigma$ and $S_{12}$ are performed simultaneously, by exploiting the current conservation relation $S_\Sigma = S_{11} + S_{22} + 2S_{12}$.

We first check that $S_\Sigma$ reflects the charge $e$ of injected electrons (Fig. 3b). This is attested by the good quantitative agreement, without any fit parameter, between the data (symbols) and the shot noise prediction for electrons (orange lines) given by[48]:

$$S_\Sigma = 2\frac{e^2}{h}\sum_{i=L,R}\tau_i(1 - \tau_i)eV_s\left[\coth\left(\frac{eV_s}{2k_BT}\right) - \frac{2k_BT}{eV_s}\right], \tag{3}$$

with $T = 11$ mK and $\tau_{L(R)}$ the measured dc transmission of the left (right) source shown in Fig. 3a.

We then focus on the cross-correlation investigation of anyonic behavior. As shown in Fig. 3c, $S_{12} \approx 0$ at low bias, up to $S_\Sigma \approx 6 \times 10^{-29}$ A² Hz⁻¹ corresponding to $|V_s| \approx 45$ µV. This $P \approx 0$ signals a trivial mutual statistics, which is expected in the low bias regime where the injected electron is not fractionalized at the analyzer. Then, at $|V_s| \gtrsim 45$ µV where the fractionalization takes place according to $f(\varepsilon)$ spectroscopy, $S_{12}$ turns negative with a slope of $P \simeq -0.38/-0.56$ (green solid/dashed line) for $\tau_s = 0.05/0.95$ respectively. The clear negative signal with a fixed slope constitutes a strong qualitative marker of non-trivial mutual braiding statistics, as further discussed below.

The relationship between negative cross-correlations and anyonic mutual statistics in the dilute limit of small $\tau_s$ (or, symmetrically, small $1 - \tau_s$) is most clearly established in a perturbative analysis along the lines of Morel et al.[23]. For $\tau_s \ll 1$ and at long distances from the source, we find (see Supplementary Information):

$$P \simeq \frac{\sin^2\theta}{\theta^2}\ln\tau_s, \tag{4}$$

with $2\theta$ the mutual braiding (double exchange) phase. For quasiparticles of charges $q$ and $q'$ along an integer quantum Hall channel, theory predicts $\theta = \pi qq'/e^2$ (see, e.g., Ref. 22). In the present case of a braiding between incident fractional charges $e/2$ and spontaneously generated electron-hole pairs, we thus have $\theta = \pi/2$ and $P \simeq \frac{4}{\pi^2}\ln\tau_s$ (see also Ref. 28 for the same prediction, but without the explicit connection to the fractional mutual statistics). Note that in the present integer quantum Hall implementation, the relationship between $P$ and $\theta$ is not complicated by additional parameters, such as the fractional quasiparticles' scaling dimension and topological spin that both come into play in the fractional quantum Hall regime[14,15,41,42]. However, achieving $P \propto \ln\tau_s$ requires large $|\ln\tau_s|$ and thus exponentially small $\tau_s$, which complicates a quantitative comparison of experimental data with Eq. (4). Accordingly, injecting $\tau_s = 0.05$ ($\ln\tau_s \simeq -3$) into Eq. (4) gives a slope $P \simeq -1.2$, substantially more negative than the observations.

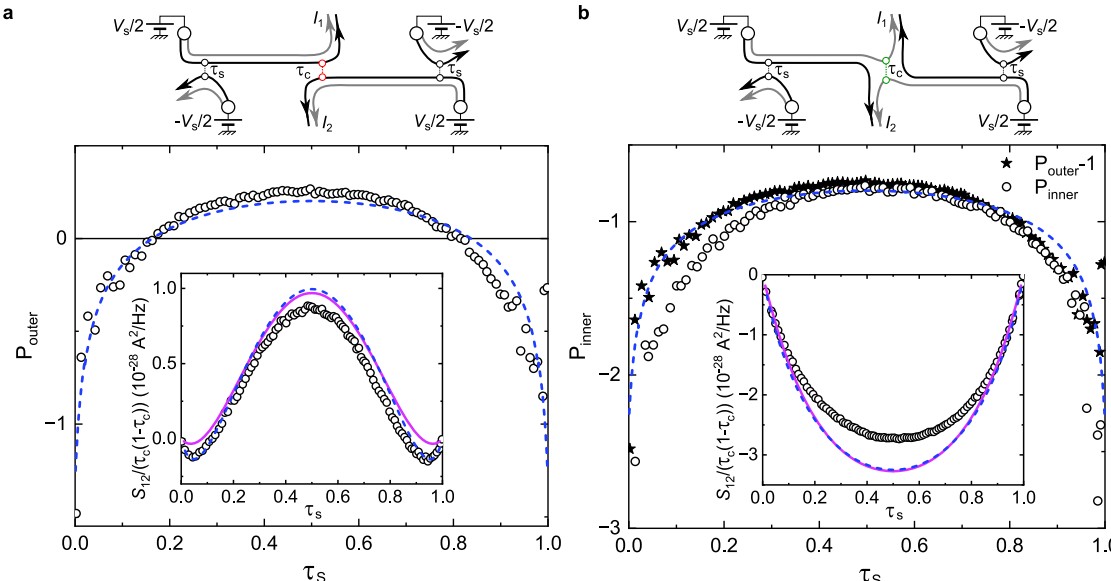

**Fig. 4 | Cross-correlations vs dilution of symmetric beams.** Main panels and insets show, respectively, the generalized Fano factor $P$ and the renormalized cross-correlations $S_{12}(V_s = 70 \ \mu V)/(\tau_c(1-\tau_c))$ vs the outer edge channel transmission $\tau_s$ of the symmetric source QPCs. Symbols are data points. Blue lines are high bias/long $\delta t$ predictions. Purple lines are $S_{12}(V_s = 70 \ \mu V)/(\tau_c(1-\tau_c))$ predictions at $\delta t = 64$ ps. **a** The cross-correlation signal and corresponding $P_{outer}$ (open circles) are measured by partially transmitting at the central QPC ($\tau_c \approx 0.5$) the same outer edge channel (black) where electrons are tunneling at the sources (see schematics). This is the standard 'collider' configuration. **b** The cross-correlation signal and corresponding $P_{inner}$ are obtained by setting the central QPC to partially transmit ($\tau_c \approx 0.5$) the inner edge channel (grey), whereas electrons are tunneling into the outer edge channel at the sources (see schematic). In this particular configuration, the source shot noise does not directly contribute to the cross-correlation signal. Filled symbols in the main panel display $P_{outer} - 1$, with $P_{outer}$ the data in **a** and $-1$ corresponding to the subtraction of the source shot noise.

A better data-theory agreement can be obtained with a non-perturbative treatment of the sources. Indeed, the predicted slope in the high bias/long distance limit at $\tau_s = 0.05$ is $P \simeq -0.35$ (see Eq. (52) in Supplementary Information), close to one of the observed values $P \simeq -0.38$. Although we expect the same cross-correlations and Fano factor for $\tau_s = 0.05$ and $0.95$, the measured value at $0.95$ of $P \simeq -0.56$ (green dashed line) is somewhat higher, and also deviates more from the theory prediction (purple dashed line). We discuss possible reasons, such as increased inter-channel tunneling, in the Supplementary Information. The full finite bias/finite distance predictions (purple lines in Fig. 3c) also reproduce the overall shape of the measurements, although with a noticeable horizontal shift (see Supplementary Information for a discussion of possible theoretical limitations). Further evidence of the underlying anyonic mechanism is provided from the effect of the dilution of the quasiparticle beam.

## Cross-correlations vs beam dilution

Here we explore the effect of dilution by sweeping the transmission across the two symmetric sources over the full range $\tau_s \in [0, 1]$, and we also extend our investigation to the inner edge channel where the electron fractionalization results in two pulses carrying opposite charges $\pm e/2$ (see Fig. 1a).

Let us first consider the previous/standard configuration, with source QPCs and analyzer QPC set to partially transmit the same, outer, channel. Away from the dilute limits, $P$ and $S_{12}$ show a change of sign (see Fig. 4a). This results from an increasing importance of the positive contribution from the noise generated at the sources[50] with respect to the noise generated at the analyzer involving the emergence of mechanisms other than time braiding (such as collisions)[23,41]. In addition we see that the data are symmetric around $\tau_s = 0.5$, due to the unchanging electron nature of tunneling particles into the IQH edges (with small deviations possibly from inter-channel tunneling, as previously mentioned). This is in contrast with the fractional quantum Hall regime where the nature of the tunneling quasiparticles changes[51] between $\tau_s \ll 1$ and $1 - \tau_s \ll 1$. The agreement with theory observed for $P(\tau_s)$, and more specifically for the $\tau_s$ dependence in the dilute limits $\tau_s \ll 1$ and $1 - \tau_s \ll 1$, further establishes the experimental cross-correlation signature of fractional mutual statistics. Note that the less precise agreement with the full $S_{12}$ signal is reminiscent of the horizontal shift of the negative slope in Fig. 3.

We then consider in Fig. 4b the alternative configuration, where the analyzer is set to $\tau_c \simeq 0.5$ for the inner edge channel (the outer edge channel, where electrons are injected at the sources, being fully transmitted, see schematics). In that case, the cross-correlation signal is always negative due to charge conservation. The positive contribution $\tau_c(1 - \tau_c)S_\Sigma$ from the partition of the current noise generated at the sources is absent since the electron tunneling at the sources does not take place in the probed inner edge channel. In the strong coupling limit where fractional charges of identical amplitude $e/2$ propagate on both inner and outer channel, the same unconventional braiding is expected to have the same cross-correlation consequences. The absence of source noise then simply results in an offset: $P_{inner} = P_{outer} - 1$ ($S_{12}^{inner} = S_{12}^{outer} - \tau_c(1 - \tau_c)S_\Sigma$), where the label inner/outer indicates the channel probed at the analyzer. For a direct comparison, $P_{outer} - 1$ is also shown (filled stars). The agreement between the two data sets provides an additional, direct confirmation that the device is in the strong coupling regime. It also experimentally establishes the robust contribution from the source noise to $S_{12}$, allowing to distinguish it from the effect of time-braiding.

## Discussion

Edge excitations are not characterized by topologically protected quantum numbers, in contrast to the quantum Hall bulk quasiparticles. Along the integer and fractional quantum Hall edges, the charge and the quantum statistics of such excitations can be varied continuously. In the present work, we exploit this property to form dilute beams of

fractional charges that behave like anyons. This is achieved by using QPCs as electron sources in combination with the intrinsic Coulomb interaction between copropagating integer quantum Hall channels. We establish their fractional quantum statistics by the emergence of negative current cross-correlations between the two outputs of a downstream analyzer QPC, similar to previous observations in the fractional quantum Hall regime. By contrast, when applying sufficiently low source bias voltages such that the tunneling electrons do not fractionalize, the absence of a cross-correlation signal coincides with their fermionic character.

We believe that the demonstrated integer quantum Hall platform opens a promising practical path to explore the emerging field of anyon quantum optics[52]. Advanced and time-resolved quantum manipulations of anyons are made possible by the large quantum coherence along the integer quantum Hall edge and the robustness of the incompressible bulk. By tailoring single-quasiparticle wave-packets, for example with driven ohmic contacts, a vast range of fractional anyons of arbitrary exchange phase becomes available along the integer quantum Hall edges, well beyond the odd fractions of $\pi$ of Laughlin quasiparticles encountered in the fractional quantum Hall regime.

## Methods

### Sample fabrication

The device, shown in Fig. 1c of the Main text, is patterned in an AlGaAs/GaAs heterostructure forming a two-dimensional electron gas (2DEG) buried 95 nm below the surface. The 2DEG has a mobility of $2.5 \times 10^6$ cm$^2$ V$^{-1}$ s$^{-1}$ and a density of $2.5 \times 10^{11}$ cm$^{-2}$. It was nanofabricated following five standard e-beam lithography steps:

1. Ti-Au alignment marks are first deposited through a PMMA mask.
2. The mesa is defined by using a ma-N 2403 protection mask and by wet-etching the unprotected parts in a solution of H$_3$PO$_4$/H$_2$O$_2$/H$_2$O over a depth of ~100 nm.
3. The ohmic contacts allowing an electrical connection with the buried 2DEG are realized by the successive depositions of Ni (10 nm) - Au (10 nm) - Ge (90 nm) - Ni (20 nm) - Au (170 nm) - Ni (10 nm) through a PMMA mask, followed by a 440 °C annealing for 50 s.
4. The split gates controlling the QPCs consist in 40 nm of aluminium deposited through a PMMA mask.
5. Finally, we deposit thick Cr-Au bonding ports and large-scale interconnects through a PMMA mask.

The nominal tip-to-tip distance of the Al split gates used to define the QPCs is 150 nm.

### Measurement setup

The sample is installed in a cryofree dilution refrigerator with important filtering and thermalization of the electrical lines, and immersed in a perpendicular magnetic field $B = 5.2$ T, which corresponds to the middle of the $\nu = 2$ plateau. Cold $RC$ filters are mounted near the device: 200 kΩ · 100 nF on the lines controlling the split gates, 10 kΩ · 100 nF on the injection lines and 10 kΩ · 1 nF on the low frequency measurement lines.

Lock-in measurements are made at frequencies below 25 Hz, using an ac modulation of rms amplitude below $k_B T/e$. We calculate the dc currents and QPC transmissions by integrating the corresponding lock-in signal vs the source bias voltage (see the following and ref. 53 for details).

The auto- and cross-correlations of the currents $I_1$ and $I_2$ (Fig. 1c) are measured with home-made cryogenic amplifiers[54] around 0.86 MHz, the resonant frequency of the two identical tank circuits along the two amplification chains. The measurements are performed by integrating the signal over the bandwidth of [0.78, 0.92] MHz. The

measurement setup is detailed in the supplemental material of ref. 55.

### Thermometry

The electron temperature in the sample is measured using the robust linear dependence of the thermal noise $S(T) \propto T$. At $T > 40$ mK, we use the (equilibrium) thermal noise plotted versus the temperature readout by the calibrated RuO$_2$ thermometer. The linearity is a confirmation of the electron thermalization and of the thermometer calibration. The quantitative value of the slope provides us with the gain of the full noise amplification chain, as detailed in the next section. To determine the temperature in the $T < 40$ mK range, we measure the thermal noise and determine the corresponding temperature by linearly extrapolating from the $S(T > 40$ mK) data. The values of $T$ obtained using the two amplification chains are found to be consistent. We also check that $T$ corresponds to the temperature obtained from standard shot noise measurements performed individually on each QPC ahead of and during each measurement. A 1 mK higher shot noise temperature is specifically associated with the top-right ohmic contact feeding the right source, and attributed to noise from the corresponding connecting line.

### Calibration of the noise amplification chain

For each noise amplification chain $i \in \{1, 2\}$, the gain factor $G_i^{\text{eff}}$ between current noise spectral density and raw measurements needs to be calibrated. From the slopes $s_1$ and $s_2$ of $S(T > 40$ mK) measured, respectively, for the amplification chain 1 and 2 (see Thermometry), and the robust fluctuation-dissipation relation $S(T) = 4k_B T \, \text{Re}[Z]$ with $Z$ the frequency dependent impedance of the tank circuit in parallel with the sample, we get:

$$G_{1(2)}^{\text{eff}} = \frac{s_{1(2)}}{4k_B(1/R_{\text{tk}}^{1(2)} + 1/R_{\text{H}})}, \tag{5}$$

with $R_{\text{H}} = h/2e^2$ the Hall resistance of the sample, and $R_{\text{tk}}^{1(2)} = 150$ kΩ (153 kΩ) the separately obtained effective parallel resistance due to the dissipation in the tank circuit connected to the same port. With $G_1^{\text{eff}}$ and $G_2^{\text{eff}}$ given by the above relation, the gain for the cross-correlation signal reads $G_{12}^{\text{eff}} = \sqrt{G_1^{\text{eff}} G_2^{\text{eff}}}$ thanks to the good match of the two resonators. For more details see ref. 53.

### Differential (ac) and integral (dc) transmission

**Source transmissions.** The transmissions of the left and right source QPC $\tau_{\text{L,R}}$ are defined as the ratio between the dc current transmitted across the QPC

$$I_{\text{L,R}} = \int_0^{V_{3,4}} \frac{\partial(I_1 + I_2)}{\partial V_{3,4}} dV_{3,4} \tag{6}$$

and the dc current $e^2 V_{3,4}/h$ incident on one side of the QPC for the considered outer edge channel (i.e., half of the total injected current), yielding:

$$\tau_{\text{L,R}} = \frac{2h}{e^2} \frac{I_{\text{L,R}}}{V_{3,4}}. \tag{7}$$

In the specific case where the sources are set to partial transmission of the inner channel (case of Fig. 7 in the Supplementary Information), source transmissions of the inner channel can similarly be defined as

$$\tau_{\text{L,R}}^{\text{in}} = \frac{2h}{e^2 V_{1,2}} \int_0^{V_{1,2}} \left( \frac{e^2}{h} - \frac{\partial(I_1 + I_2)}{\partial V_{1,2}} \right) dV_{1,2}. \tag{8}$$

**Analyzer transmission.** Whereas above we use dc transmissions for the sources, in all measurements the relevant transmission of the analyzer is the ac transmission, obtained as

$$\tau_c = \left(\frac{\partial I_2}{\partial V_3}\right) \Big/ \left(\frac{\partial (I_1 + I_2)}{\partial V_3}\right). \tag{9}$$

In particular, there is no dc bias of the analyzer in the symmetric source configuration probing the quantum statistics.

## Data availability
Plotted data, raw data, data analysis code and numerical code used to calculate the full theoretical predictions have been deposited in Zenodo under the accession code: https://doi.org/10.5281/zenodo.10492057.

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

## Acknowledgements

We would like to thank P. Degiovanni and J. T. Chalker for discussions. This work was supported by the European Research Council (ERC-2020-SyG-951451), the French National Research Agency (ANR-16-CE30-0010-01 and ANR-18-CE47-0014-01), the French RENATECH network and DIM NANO-K. D.K. acknowledges support from Labex MME-DII grant ANR11-LBX-0023, and funding under The Paris Seine Initiative EMER-GENCE programme 2019.

## Author contributions

P.G., C.P. and I.P. performed the experiment and analyzed the data with inputs from A.Aa., A.An., D.K. and F.P.; D.K. and C.M. developed the theory; D.K. wrote the code for numerical predictions; A.C. and U.G. grew the 2DEG; P.G and A.Aa. fabricated the sample with inputs from A.An.; Y.J. fabricated the HEMTs used in the cryogenic amplifiers for noise measurements; I.P. and P.G. wrote the manuscript with contributions from all authors; A.An. and F.P. led the project.

## Competing interests

The authors declare no competing interests.
