## [Peer Review File · Nature Communications]

REVIEWER COMMENTS

Reviewer #1 (Remarks to the Author):

I find this paper very interesting. It shows the physics similar to that of fractional statistics on an integer quantum Hall edge. I see the results as sufficiently interesting to warrant publication in Nature Communications.

At the same time, there is a fundamental difference between the physics observed in this paper and actual anyonic statistics. Fractional statistics in FQHE is a manifestation of topological order. It is protected by a bulk energy gap, and statistical phases assume quantized values, though long-range forces may add a non-universal correction. The set and statistics of the anyons are not continuously tunable. This is rather different from what happens on an integer QHE edge. There is no gap and the charge of an anyon-like object is not generally quantized. The existence of a non-trivial effective braiding phase $2\pi\theta$ of an electron or hole and a fractional object demonstrates this well. Indeed, in FQHE we normally expect that all anyons braid trivially with electrons.

The above point puts the results in the proper context and perhaps it would be useful to discuss this point in the manuscript. At the same time, this does not subtract from the interest of the results.

The paper is well written. My primary concern consists in an incomplete coverage of the theoretical side by this paper. The paper combines an experimental paper with a lengthy theoretical Supplementary Information. While Supplementary Information is long, it appears to this referee that it is incomplete, that is, some theoretical derivations needed for the data interpretation are not included and will be published in a subsequent paper. Given the length of the relevant theory, it is a good idea to have theoretical results published separately. However, they should be available before the publication of this paper. I am not sure how to handle this best and leave this to the authors and editors. Perhaps, a detailed theory paper should be posted on arXiv before the publication of this manuscript.

Supplementary Information discusses in detail the asymmetry with respect to τ_s . I think it is essential to address this point in the main text since this sheds light on the limitations of the approach and the data.

Other than the above comments, I do not have much criticism. The experiment appears sound to me. There are a few minor points about Supplementary Information.

First, I am somewhat uncomfortable with the discussion of gauge invariance on page 1 of Supplementary Information. Indeed, besides the voltage bias, one needs to take the potential of the gates into account. I am not sure that the gauge-invariance argument survives that, but this is a minor point.

Second, it would be helpful to clarify that Supplementary Information uses its own separate list of references. Would it perhaps be worth it to move all references to the main text?

Third, the equation $\tau=1-\tau$ between (53) and (54) implies $\tau=1/2$. This is not what the authors actually try to tell.

I am not sure what is meant by the comment that the short-range interaction model does not apply at low voltages in Section XI. Do the authors mean that the gates are inefficient in screening the long-range Coulomb interaction, so that long-range interactions must be taken into account at low energies? If this is the reason for the comment, could the authors explain inefficient screening?

In conclusion, this is an interesting paper. It can be published in Nature Communications after a modest revision. At the same time, it is desirable to have a full theory available to the readers before the publication.

Reviewer #2 (Remarks to the Author):

Anyonic statistics of elementary excitations in quantum Hall systems have attracted much attention since the first report for $e/3$ Laughlin quasiparticles in 2020. While assuming non-Abelian anyon braiding in fractional quantum Hall states as the goal, many researchers are making efforts to investigate in detail the properties of anyons that exhibit other fractional statistics. In addition to the interferometry, 'collider' experiments have revealed the fractional statistics of quasiparticles in fractional quantum Hall systems through time-domain braiding. The authors of this manuscript applied this experimental method to investigate the quantum statistics of fractionalized charges in interacting integer quantum Hall systems, which mimic the anyonic excitations in the fractional quantum Hall regime.

Interacting integer quantum Hall edge states can host fractional charge excitations that mimic fractional anyons through charge fragmentation.

In this experiment, electrons are injected into the interacting channels through a quantum point contact (QPC) by tunneling effect. These electrons are fractionalized into fast and slow excitations, which possess fractional charges in each channel, after propagation of some distance, to reach an analyzer QPC. The statistical nature of the fractional charges is evaluated by measuring the current noise generated at the analyzer. The authors first confirmed the fractionalization process by performing energy spectroscopy. Then, they examined the anyonic statistics by analyzing the cross-correlation of the output current noise generated in the 'collider' setup. The quantitative analysis on the negative cross-correlation for diluted beam injections clearly demonstrates the fractional statistics as the first observation of the anyonic nature in integer quantum Hall systems.

This well-written manuscript deserves publication in Nature Communications. The measurements and analyses are of a high level of sophistication. The experimental results are novel and in line with current trends in condensed matter physics. I recommend publishing this manuscript after the authors address a few comments listed below.

1. On page 3, right column, the authors describe that only the data for the source bias 70 μV (Fig. 2c) agrees well with the long-distance prediction so that the separation of the wave packets is established between 35 μV and 70 μV . I think the flow of logic is unclear in this part. The authors should explain how they determined the bias range over which the separation is achieved.

2. Section IV of supplementary information: The authors discuss that the mere presence of negative cross-correlations cannot be directly attributed to the anyonic exchange phase and that they have validated the non-perturbative approach by complementary measurements in the main text. However, the main text does not mention which specific experimental results rule out the phenomenological fermionic theory. I recommend adding explanations for this point in the main text.

3. (Minor comment) In the last commutation relation in Equation (7), the last delta has lost its subscript.

Reviewer #3 (Remarks to the Author):

In the integer quantum Hall effect regime, the paper describes the ‘time braiding’ of induced wave-packets, charged $+e/2$ and $-e/2$ (dipoles, proposed in Ref. 26). The packets are produced in filling factor 2, with the outer edge mode being partitioned in a quantum point contact (QPC), and the above-mentioned dipoles are induced due to Coloumb interactions in the inner edge mode. The latter packets are time braided in a second QPC, where electron-hole excitations braid around the arriving fractional packets. This experiment mimics the previous works performed in the fractional regime, and its uniqueness is demonstrating ‘fractional braiding’ in the integer regime.

I suggest accepting the paper but after making the following changes (additions):

Reference 26 (mentioned above) should be more respected and highlighted instead of buried with other references.

A previous work that produced and studied these fractional packets should be referenced by H. Inoue et al. PRL 112, 166801 (2014). The last pp (summary and outlook) are poorly written, and a reader (not versed with the full text) would be lost; it should be rewritten in a more friendly and tutorial way.

We thank the Reviewers for their positive and useful comments. Below is our point-by-point reply.

Reviewer #1:

I find this paper very interesting. It shows the physics similar to that of fractional statistics on an integer quantum Hall edge. I see the results as sufficiently interesting to warrant publication in Nature Communications.

We thank the Reviewer for the positive and useful comments.

At the same time, there is a fundamental difference between the physics observed in this paper and actual anyonic statistics. Fractional statistics in FQHE is a manifestation of topological order. It is protected by a bulk energy gap, and statistical phases assume quantized values, though long-range forces may add a non-universal correction. The set and statistics of the anyons are not continuously tunable. This is rather different from what happens on an integer QHE edge. There is no gap and the charge of an anyon-like object is not generally quantized. The existence of a non-trivial effective braiding phase $2\pi\theta$ of an electron or hole and a fractional object demonstrates this well. Indeed, in FQHE we normally expect that all anyons braid trivially with electrons.

The above point puts the results in the proper context and perhaps it would be useful to discuss this point in the manuscript. At the same time, this does not subtract from the interest of the results.

We agree with the Reviewer on the point that anyonic quasiparticles in the fractional quantum Hall bulk benefit from topological protection and have a quantized statistical phase and charge. Indeed, in this way they differ from the excitations we have studied in the integer edge. At the same time, these excitations are predicted to have fractional braiding statistics, which we observe through a time-braiding signature in the cross-correlations. This is the basis on which we consider them to be anyons.

We have added a paragraph in the Discussion section which clarifies this point, as well as a sentence in the introduction.

The paper is well written. My primary concern consists in an incomplete coverage of the theoretical side by this paper. The paper combines an experimental paper with a lengthy theoretical Supplementary Information. While Supplementary Information is long, it appears to this referee that it is incomplete, that is, some theoretical derivations needed for the data interpretation are not included and will be published in a subsequent paper. Given the length of the relevant theory, it is a good idea to have theoretical results published separately. However, they should be available before the publication of this paper. I am not sure how to handle this best and leave this to the authors and editors. Perhaps, a detailed theory paper should be posted on arXiv before the publication of this manuscript.

In response to the Reviewer, we have decided to at the same time clarify points in the Supplementary Information, and to aim for the deposition of a dedicated theoretical article on arXiv close to the publication of the present manuscript. We enclose a file which contains the more detailed outline of the theory, both as demonstration of work in progress, and to allow to the Reviewer to gain a better overview. At this time we have added points to the SI which make it easier to make a comparison to the experiment and to follow the code available on zenodo, which had

been used to produce the theory curves in the paper.

Supplementary Information discusses in detail the asymmetry with respect to τ_s . I think it is essential to address this point in the main text since this sheds light on the limitations of the approach and the data.

We agree with this point, and in response we have added the dataset at $\tau_s = 0.95$ to Fig. 3 in the Main text (in addition to $\tau_s = 0.05$), as well as the discussion of the experimental discrepancy between the measurements at τ_s and $1-\tau_s$.

Other than the above comments, I do not have much criticism. The experiment appears sound to me. There are a few minor points about Supplementary Information.

First, I am somewhat uncomfortable with the discussion of gauge invariance on page 1 of Supplementary Information. Indeed, besides the voltage bias, one needs to take the potential of the gates into account. I am not sure that the gauge-invariance argument survives that, but this is a minor point.

The theoretical model does not involve QPC gate voltages but directly the transmission τ across the QPCs. Indeed, in the experimental implementation, a gauge change corresponding to a global potential shift also implies a shift of the gate voltages. If not, the potential barrier at the QPC and therefore the transmission τ would not be preserved.

To clarify this sentence, we removed 'gauge invariance' and used the simpler formulation 'invariance under global potential shift'.

Second, it would be helpful to clarify that Supplementary Information uses its own separate list of references. Would it perhaps be worth it to move all references to the main text?

It is our understanding that it is not allowed by the editorial rules of the journal to put the references of the Supplementary Information into the main text.

As suggested by the Reviewer, we have added a note at the start of the Supplementary Information to point out that it has its own set of references.

Third, the equation $\tau = 1 - \tau$ between (53) and (54) implies $\tau = 1/2$. This is not what the authors actually try to tell.

We thank the Reviewer for pointing out this typo, we were referring to the $\tau \rightarrow 1 - \tau$ symmetry and have clarified this sentence in the Supplementary Information.

I am not sure what is meant by the comment that the short-range interaction model does not apply at low voltages in Section XI. Do the authors mean that the gates are inefficient in screening the long-range Coulomb interaction, so that long-range interactions must be taken into account at low energies? If this is the reason for the comment, could the authors explain inefficient screening?

We thank the referee for pointing this out. This sentence was premature (and should not have appeared in the Supplementary) as we are still trying to understand the effect of long range

interactions. It is now removed.

In conclusion, this is an interesting paper. It can be published in Nature Communications after a modest revision. At the same time, it is desirable to have a full theory available to the readers before the publication.

Reviewer #2:

Anyonic statistics of elementary excitations in quantum Hall systems have attracted much attention since the first report for $e/3$ Laughlin quasiparticles in 2020. While assuming non-Abelian anyon braiding in fractional quantum Hall states as the goal, many researchers are making efforts to investigate in detail the properties of anyons that exhibit other fractional statistics. In addition to the interferometry, 'collider' experiments have revealed the fractional statistics of quasiparticles in fractional quantum Hall systems through time-domain braiding. The authors of this manuscript applied this experimental method to investigate the quantum statistics of fractionalized charges in interacting integer quantum Hall systems, which mimic the anyonic excitations in the fractional quantum Hall regime.

Interacting integer quantum Hall edge states can host fractional charge excitations that mimic fractional anyons through charge fragmentation.

In this experiment, electrons are injected into the interacting channels through a quantum point contact (QPC) by tunneling effect. These electrons are fractionalized into fast and slow excitations, which possess fractional charges in each channel, after propagation of some distance, to reach an analyzer QPC. The statistical nature of the fractional charges is evaluated by measuring the current noise generated at the analyzer. The authors first confirmed the fractionalization process by performing energy spectroscopy. Then, they examined the anyonic statistics by analyzing the cross-correlation of the output current noise generated in the 'collider' setup. The quantitative analysis on the negative cross-correlation for diluted beam injections clearly demonstrates the fractional statistics as the first observation of the anyonic nature in integer quantum Hall systems.

This well-written manuscript deserves publication in Nature Communications. The measurements and analyses are of a high level of sophistication. The experimental results are novel and in line with current trends in condensed matter physics. I recommend publishing this manuscript after the authors address a few comments listed below.

1. On page 3, right column, the authors describe that only the data for the source bias 70 μV (Fig. 2c) agrees well with the long-distance prediction so that the separation of the wave packets is established between 35 μV and 70 μV . I think the flow of logic is unclear in this part. The authors should explain how they determined the bias range over which the separation is achieved.

We thank the Reviewer for the positive review. We have modified the manuscript to clarify our arguments.

2. Section IV of supplementary information: The authors discuss that the mere presence of negative cross-correlations cannot be directly attributed to the anyonic exchange phase and that they have

validated the non-perturbative approach by complementary measurements in the main text. However, the main text does not mention which specific experimental results rule out the phenomenological fermionic theory. I recommend adding explanations for this point in the main text.

We find that the observed non-trivial evolution of the energy distribution function with voltage bias closely matches the theoretical predictions obtained from the non-perturbative approach, with only one fitting parameter, which strongly supports this model.

Note that we did not directly rule out the phenomenological fermionic theory because this would require one to compare the observations with specific predictions regarding the evolution of the energy distribution whereas this theory does not allow one to make this kind of predictions (this would require to introduce a choice for the rate of inelastic collisions as a function of the exchanged energy).

We have clarified this in the Main text and added a note in the Supplementary Information section V (former section IV).

3. (Minor comment) In the last commutation relation in Equation (7), the last delta has lost its subscript.

We thank the Reviewer for pointing out this typo. We have added the missing subscript to say δ_{ss} .

Reviewer #3:

In the integer quantum Hall effect regime, the paper describes the 'time braiding' of induced wave-packets, charged $+e/2$ and $-e/2$ (dipoles, proposed in Ref. 26). The packets are produced in filling factor 2, with the outer edge mode being partitioned in a quantum point contact (QPC), and the above-mentioned dipoles are induced due to Coloumb interactions in the inner edge mode. The latter packets are time braided in a second QPC, where electron-hole excitations braid around the arriving fractional packets. This experiment mimics the previous works performed in the fractional regime, and its uniqueness is demonstrating 'fractional braiding' in the integer regime. I suggest accepting the paper but after making the following changes (additions):

Reference 26 (mentioned above) should be more respected and highlighted instead of buried with other references.

We thank the Reviewer for the positive overall assessment and for this remark. We have added a sentence in the Main text which singles out Reference 26 as a theoretical proposal for the generation of fractional charges along integer quantum Hall edges. Ref. 26 is cited three times in the Main text when discussing various facets of charge fractionalization.

A previous work that produced and studied these fractional packets should be referenced by H. Inoue et al. PRL 112, 166801 (2014).

We have now added a citation to this reference.

The last pp (summary and outlook) are poorly written, and a reader (not versed with the full text) would be lost; it should be rewritten in a more friendly and tutorial way.

In response to the Reviewer we have improved the summary.

List of changes:

Main text:

- Please see the annotated version, the changes are given in red
- We have added reference [36] Inoue, H. et al. Charge fractionalization in the integer quantum Hall effect. Phys. Rev. Lett. 112, 166801 (2014)
- We have added the dataset at $\tau_s = 0.95$ and the corresponding theory prediction to Figure 3 following the remarks of Reviewer 1

Supplementary Information:

- In Section I “Non-perturbative theory” we made changes to clarify the theoretical points, and we added elements to specify how the comparison to the experiment is conducted in all configurations. To that end we added subsections “Collider configuration/Finite temperature expressions” and “Electron energy distribution configuration”
- We moved the figures from Extended Data into the Supplementary Information, and they make up Section VI in the distribution configuration (previously Extended Data Figs. 1,2,3) and Section IX in the collider configuration (previously Extended Data Fig. 4)
- We added two sentences at the end of Section V (former Section IV) as comment on the applicability of the phenomenological approach
- We made small adjustments in Section VII to be coherent with the change of Figure 3 in the Main text
- We removed a sentence from Section XIII in response to the comment of Reviewer 1 on the applicability of short-range interactions.

The changes in the Supplementary Information are not annotated.